# cAMP Signaling in Cancer: A PKA-CREB and EPAC-Centric Approach

**DOI:** 10.3390/cells11132020

**Published:** 2022-06-24

**Authors:** Muhammad Bilal Ahmed, Abdullah A. A. Alghamdi, Salman Ul Islam, Joon-Seok Lee, Young-Sup Lee

**Affiliations:** 1BK21 FOUR KNU Creative BioResearch Group, School of Life Sciences, College of Natural Sciences, Kyungpook National University, Daegu 41566, Korea; mbilalknu@gmail.com (M.B.A.); joonseok74@naver.com (J.-S.L.); 2Department of Biology, Faculty of Science, Albaha University, Albaha 65779, Saudi Arabia; aaa.alghamdi@bu.edu.sa; 3Department of Pharmacy, Cecos University, Peshawar, Street 1, Sector F 5 Phase 6 Hayatabad, Peshawar 25000, Pakistan; salmanulislam@cecos.edu.pk

**Keywords:** cAMP, PKA, CREB, EPAC, tumor cell

## Abstract

Cancer is one of the most common causes of death globally. Despite extensive research and considerable advances in cancer therapy, the fundamentals of the disease remain unclear. Understanding the key signaling mechanisms that cause cancer cell malignancy may help to uncover new pharmaco-targets. Cyclic adenosine monophosphate (cAMP) regulates various biological functions, including those in malignant cells. Understanding intracellular second messenger pathways is crucial for identifying downstream proteins involved in cancer growth and development. cAMP regulates cell signaling and a variety of physiological and pathological activities. There may be an impact on gene transcription from protein kinase A (PKA) as well as its downstream effectors, such as cAMP response element-binding protein (CREB). The position of CREB downstream of numerous growth signaling pathways implies its oncogenic potential in tumor cells. Tumor growth is associated with increased CREB expression and activation. PKA can be used as both an onco-drug target and a biomarker to find, identify, and stage tumors. Exploring cAMP effectors and their downstream pathways in cancer has become easier using exchange protein directly activated by cAMP (EPAC) modulators. This signaling system may inhibit or accelerate tumor growth depending on the tumor and its environment. As cAMP and its effectors are critical for cancer development, targeting them may be a useful cancer treatment strategy. Moreover, by reviewing the material from a distinct viewpoint, this review aims to give a knowledge of the impact of the cAMP signaling pathway and the related effectors on cancer incidence and development. These innovative insights seek to encourage the development of novel treatment techniques and new approaches.

## 1. Introduction

The growth of cells is supported by a variety of signaling pathways. Tumorigenesis is triggered by improperly activated or suppressed signal transduction pathways. Post-translational processes including phosphorylation, ubiquitination, methylation, and acetylation are all involved in cell signaling control. Protein kinases and phosphatases are often abnormally or uncontrollably activated in cancers, making them prime candidates for molecularly targeted tumor therapies.

Signaling pathways are often triggered by signal molecules found on membranes or within cells, such as those produced by growth factors, hormones, or ions (Figure 1). To carry out this process, a variety of feedback mechanisms and intracellular chemicals, known as second messengers, are used. These include enzymes that control the release of these molecules as well as the actual calcium ions [1]. Second messenger cAMP was discovered in the late 1950s [2]. ATP and phosphodiesterase (PDE) are the two enzymes that manufacture it from adenylyl cyclase and degrade it to adenosine 5-monophosphate [1,3]. cAMP has the potential to affect a wide range of physiological processes, including metabolism, channel activation, cell proliferation and differentiation, gene expression, and cell death [4,5,6,7,8], because both internal and external impulses may impact the efficiency of cAMP synthesis and degradation [9].

Many intracellular signaling pathways are affected by cAMP, including Ca^2+^-mediated [11] and cytokine pathways [12]. It also works closely with Ras-mediated mitogen-activated protein kinase (MAPK), which controls cell growth [13]. cAMP acts via targeting downstream effectors such as PKA, EPACs, and cyclic nucleotide-gated ion channels (CNGC). PKA is a key effector that may phosphorylate CREB. CREB is a transcriptional cofactor that triggers a variety of transcriptional cascades and target gene expression [14]. PKAs are abundant inside cells and control many different processes. Their targets are determined by their placement inside the cell. This is performed by anchoring them in specific places in macromolecular complexes and allowing them to make specific subunits. The tetramer PKA consists of two regulatory and two inactive catalytic subunits [4,15,16]. The catalytic subunits are activated, phosphorylate a range of target proteins, and change their biological functions when each regulatory subunit binds to two molecules of cAMP. One of the four regulatory subunits (RIα/β and RIIα/β) was shown to be differentially expressed in various cells [15,16]. PKA’s four regulatory isoforms have similar structural features, while their biochemical properties vary substantially [17]. Three catalytic subunits (Cα/β/γ) may be combined with the regulatory subunits to create enzymes with a variety of biochemical properties. Under healthy and pathological situations, the PKA holoenzyme’s composition and intracellular location may be changed, resulting in a variety of effects [18]. Many PKA anchoring proteins (AKAPs) segregate PKA molecules into subcellular microdomains so that extracellular signaling receptors can only activate a small fraction of the PKA molecules within each microdomain. This could explain why different agonists elicit different physiological responses in the same cell [12,19]. PKA may dock and concentrate near critical targets to phosphorylate certain proteins more often when AKAPs bind to cytoskeletal proteins or organelles and engage regulatory subunits of PKA [20,21].

In 1998, two different research teams [8,16] made the discovery of EPAC [22,23]. EPAC proteins are multi-domain polypeptides made up of a C-terminal catalytic region and an N-terminal regulatory region (Figure 2). The catalytic region consists of a Ras-exchange motif (REM) domain, a Ras-association (RA) domain, and a cell division cycle 25 homology domain (Cdc25-HD) [24]. While Cdc25-HD is responsible for the guanine nucleotide exchange factor (GEF) activity of EPAC [25], the REM and RA domains play roles in stabilizing the active conformation of EPAC and targeting EPAC to the membrane, respectively [26,27]. The regulatory region is made up of a cAMP-nucleotide binding-B domain. (CNBD-B) and a disheveled/Egl-10/pleckstrin (DEP) domain [24]. As its name implies, the CNBD-B acts as the binding site for cAMP, whereas the DEP domain plays a role in translocating EPAC from the cytosol to the plasma membrane [28]. EPAC adopts an autoinhibitory conformation where the interaction between the regulatory CNBD and the catalytic Cdc25-HD locks EPAC in an inactive state and hinders the accessibility of Rap to the catalytic domain [22]. The binding of cAMP to CNBD induces a conformational change, releasing the autoinhibition and exposing Cdc25-HD for the Rap [28] (Figure 2).

AC generates cAMP (Green dots) from ATP after being activated by the Gα component of the Gs protein. When cAMP binds to the CNBD-B inside EPAC’s regulatory domain, it causes a conformational shift that is essential to relieve the autoinhibitory effect. Rap1/2 is then permitted to bind to Cdc25-HD, where it is activated by EPAC’s GEF activity. cAMP-EPAC proteins activate the structure of EPAC. EPAC is made up of two domains: a catalytic region with three domains and a regulatory region with two domains. The catalytic region’s domains are REM, RA, and Cdc25-HD, whereas the regulatory region’s domains are DEP and CNBD-B. EPAC1 and EPAC2 have structural variations in their regulatory areas. CNBD-A and CNBD-B are the two cAMP-binding domains found in EPAC2A. In EPAC2C, the DEF domain is absent. AC: adenylyl cyclase; Ras-associated protein 1/2 (Rap1/2); GPCR: G-protein-coupled receptor; Ras exchange motif (REM), Ras-association (RA), cell division cycle 25 homology domain (Cdc25-HD), and the domains of the regulatory region are disheveled/Egl-10/pleckstrin (DEP) and cAMP nucleotide binding-B domain (CNBD-B); guanine nucleotide exchange factor (GEF); exchange protein directly activated by cAMP (EPAC).

EPAC has been associated with various cellular processes, including proliferation, apoptosis, migration, and adhesion, despite its early discovery [26,29,30]. EPAC affects cell mitogenesis, cytoskeletal remodeling, inflammation, and oxidative stress, among other signaling pathways. EPAC and PKA, in general, govern cellular processes either alone or together. EPAC and PKA activation are required for cAMP-induced endothelial cell–cell junction stability, for example. EPAC/Rap1 signaling does not improve endothelium integrity, but PKA does [31]. Conversely, EPAC and PKA may work together to control thyroid cell growth [32]. The EPAC–cancer relationship is currently being investigated, even though the cAMP/PKA signaling pathway is well-known in cancer formation [1,29]. EPAC, a newly discovered cAMP effector, has a dual function in cancer, encouraging or inhibiting cancer formation and progression. As a result, EPAC might be used as a cancer therapy target [30,33].

According to the tumor and its surrounding environment, the cAMP pathway and its downstream effectors may either inhibit or promote cancer [1]. Given how important cAMP is, interfering with the way it transmits impulses may cause metabolic disorders and diseases in many organisms. Cancer is one of the diseases that causes lesions to grow. It is caused by cells that divide and change in ways that are not normal. This is most often seen as apoptosis being stopped or not enough cells dying. Recent research in the field of cancer has shown that there are many logical ways to stop tumors from forming [34,35,36]. Many scientists have turned their attention to the universal small organic molecule cAMP, which affects important cellular signaling transduction and controls physiological metabolic processes [37,38,39,40]. Because of this, cAMP has become a great target for modern treatments for cancer and tumor diseases as well as for making new drugs and stopping bacteria from spreading. There are many ways to target the cAMP signaling system, such as synthases, hydrolyses, and downstream effector proteins and RNAs. However, the exact molecular mechanism of how cAMP causes or protects against cancer is still not well-understood, and the clinical effect of treatment is also not well-understood.

## 2. The cAMP-PKA Pathway’s Role in the Growth of Various Tumors

cAMP is a small molecule that mediates intracellular signal transduction as a second messenger [41]. Early research suggested that cAMP signaling is predominantly dependent on PKA activation [42]. Four additional major mammalian effector protein families, including EPACs 1 and 2, the CNG channels, proteins with Popeye domains, and a cyclic nucleotide receptor implicated in sperm function (CHRIS), have also been demonstrated to be direct targets of cAMP [25]. PKA and EPAC have received a lot of attention in cancer research [43]. Cancer cells, such as glioblastoma, ovarian cancer, colorectal cancer, breast cancer, and pituitary tumors, use the cAMP/PKA signaling pathway for invasion, migration, adhesion, clonal development, and other malignant characteristics [44,45,46,47,48]. PKA phosphorylation of vasodilator-stimulated phosphoprotein (VASP) increases esophageal squamous cell invasion and metastasis [49]. As a result, PKA seems to be critical for cancer transformation. Most of the cAMP signaling research in cancer has been focused on identifying potential therapeutic targets [50]. Scientific studies using visualizations and statistical analyses are also being conducted on the cAMP signaling pathway to identify emerging patterns and hot spots in oncology research [51]. Autophagy, which is facilitated by cAMP-induced Poly [ADP-ribose] polymerase 1 (PARP1) activation, may help cure acute lymphoblastic leukemia [52]. Cell-type variations in DNA repair regulation by cAMP signaling after irradiation may be used to examine how cAMP signaling plays diverse roles in DNA repair [53]. Similarly, there may be a connection between the self-renewal of Schwann cell precursors and neurofibroma development through the purinergic receptor P2Y14 (P2RY14)/cAMP signaling pathway [54]. There are, however, several notable studies that concentrate on the function of cAMP in various cancer. For example, research suggests that boosting the levels of cAMP in hepatocellular carcinoma (HCC) cells may slow their development [55,56,57]. Several substrates, including CDC42-interacting protein 4 (CIP4), have been shown to be phosphorylated by PKA, promoting HCC invasion and metastasis [58]. cAMP, on the other hand, seems to have a paradoxical function in the development of HCC. Inhibiting B-cell lymphoma-extra-large (Bcl-Xl) expression, the vasoactive intestinal peptide reduced cAMP levels, CREB expression, and phospho-CREB (Ser133) phosphorylation in Huh7 cells [59]. The fact that cAMP plays such a broad role in HCC may be because it has so many different targets. Thus, whether cAMP promotes or inhibits HCC may vary according to the specific situation. For HCC development, cAMP levels may be crucial to maintaining homeostasis. Interestingly, the discovery of a PKA fusion protein that causes fibrolamellar hepatocellular carcinoma (FL-HCC), a rare type of liver cancer that affects less than 1% of people, was a real shock to the field [60]. FL-HCC is very different from most liver cancers because it affects children and young adults with no underlying medical condition. Most liver cancers affect adults with liver damage, usually caused by a viral infection or drinking too much. As mentioned previously, patients with FL-HCC were found to have an in-frame fusion of DnaJ homolog subfamily B member 1 (DNAJB1) and PKA Cα (DNAJB1-protein kinase cAMP-activated catalytic subunit alpha (PRKACA)), which led to increased PKA activity because the catalytic subunit was overexpressed [61]. However, overexpression of PRKACA does not fully recapitulate the oncogenicity of the fusion protein [62]. So far, DNAJB1-PRKACA has been found in almost 80% of FL-HCC patients through multiple studies [63]. Notably, several patients with FL-HCC who did not have the DNAJB1-PRKACA fusion protein but had a history of Carney complex and other tumors lost all of their regulatory subunit 1-alpha (Riα) protein [64]. Recent studies have shown that PKA fusion proteins may have an even bigger role than previously thought. They may also be driving oncogenes in extrahepatic cholangiocarcinoma, intraductal oncocytic papillary neoplasms (IOPNs), and intraductal papillary mucinous neoplasms (IPMNs) of the pancreas and bile duct [65,66,67]. Even though DNAJB1-PRKACA in FL-HCC clearly shows that PKA is an oncogenic driver, a larger analysis of cancer genomes showed that guanine nucleotide-binding protein alpha stimulating (GNAS) is the most frequently mutated G protein. It has mutations in over 4% of all sequenced tumors to date, and most of these are hotspot mutations [68,69]. Surprisingly, another study shows that GNAS-mutated cancers are more likely to be gastrointestinal cancers, such as colorectal adenocarcinoma (4–10%), stomach adenocarcinoma (6–10%), and pancreatic adenocarcinoma (5–12%) [10]. This is true for GPCRs and other G protein subunits (6–10%) and pancreatic adenocarcinoma (5–12%), a finding that extends to GPCRs and other G protein subunits [68,69,70]. GNAS and PKA also seem to play a big role in neuroendocrine cancers of the pancreas, prostate, liver, and lungs [62,70,71,72,73].

In glioma cells, stimulation of the cAMP pathway via type II isoforms of cyclic adenosine monophosphate (cAMP)-dependent protein kinase A (PKA RII) induces cell differentiation and death [74]. Expression of A-kinase anchoring protein 1(AKAP1), which attaches PKA to the cytoskeleton, as well as phosphodiesterase 1A (PDE1A), a cAMP-degrading enzyme, were shown to be increased in glioblastoma specimens [75], whereas the catalytic subunit of PKA was found to be decreased in high-grade gliomas [76]. An increase in cAMP levels, which reduces phosphatidylinositol 3-kinase (PI3K), also decreases neuroblastoma cell proliferation [77]. Changes in the intracellular microenvironment may reverse the interactions of secondary messenger pathways, which affect cellular processes crucial to cancer. For example, boosting cAMP levels may cause the stimulation pattern of the type I isoforms of cyclic adenosine monophosphate (cAMP)-dependent protein kinase A (PKA RI) (high affinity) subunits to shift to RII (low affinity) subunits [78]. In malignant gliomas, the intracellular cAMP content fluctuates during the cell cycle, with higher levels observed in the G0-G1 phase and lower levels during mitosis. It induces cell differentiation and death by arresting the cell cycle and changing the rate of subunit breakdown [74]. Glioblastoma cells have reduced cAMP levels and adenylyl cyclase activity compared to healthy brain tissue [79]. Glioma cells change their morphology and differentiation in response to the increased intracellular cAMP levels induced by various stimuli, while their proliferation is inhibited [80,81,82]. Changes in transcription are what make PKA work on glioma cells. cAMP-induced differentiation [83] stops the expression of some proteins, such as c-Jun, while it boosts the expression of other proteins, such as glial fibrillary acidic protein (GFAP) [84,85]. Changes in the cAMP pathway have been thought to be a possible cause of immortalization, the first step in the process that leads to cancer [84]. Medulloblastoma is a cerebellar cancer. During the development of the cerebellum, Purkinje cells release sonic hedgehog factor. This causes granule cell precursors to multiply, which is stopped by activating AC [86]. As reported earlier, when the amount of cAMP is raised, the growth rate of medulloblastoma cells slows and they start to differentiate [87,88]. Medulloblastoma development is suppressed when C-X-C motif chemokine receptor 4 (CXCR4) activity is restricted, which increases cAMP production comparable to phosphodiesterase blocking [89].

Furthermore, the cAMP–PKA signaling pathway is required for high levels of osteocalcin (OCN) and ostesialin (BSP) production in the androgen-independent prostate cancer cell line C4-2B [90]. Nude mice with prostate (PC-3 and DU145) tumors had their tumor growth halted by inhibiting PAK4 in PC-3 and DU145 cells [91]. Depressive and behavioral stress may also speed up the progression of prostate cancer by activating PKA [92,93].

During the development of cancer, normal cell activity is thrown off balance by changing the way certain proteins are made or broken down or by changing the way normal proteins work. Since PKA is involved in many different functions inside cells, it is possible that pathological processes could affect the cAMP/PKA pathway. In fact, several different sets of data show that the cAMP/PKA signaling pathway is changed in different cancers and could be used to diagnose or treat cancer. Many tumors have been shown to have altered signaling pathways, including the cAMP/PKA signaling system, which might be used to diagnose and treat cancer.

## 3. Involvement of CREB in Tumor Growth

In addition to its normal function, CREB is linked to the change of healthy cells into cancerous ones. Its constant and frequent activation is enough to turn normal cells into tumor cells. This happens when the receptor tyrosine kinase (RTK), cytokine/JAK/STAT signaling pathways, and downstream signaling pathways are all activated in an abnormal way (Figure 3). CREB overexpression has been found in solid tumors such as non-small-cell lung carcinoma (NSCLC), glioblastoma, breast carcinoma, melanoma, and diffuse malignant mesothelioma [94,95,96,97,98,99,100,101,102,103,104,105] as well as hematological malignancies [106,107,108,109]. There has also been increased cell division, less apoptosis, more blood vessel growth, and differentiation caused by radiation [110]. Furthermore, CREB overexpression has been linked to clinicopathological criteria such as tumor stage, grade, metastasis, increased recurrence, poor prognosis, and decreased tumor patient survival [100,111,112,113,114]. This was because CREB overexpression caused the expression of CREB target genes with CRE elements in their promoters to increase. Using chromatin immunoprecipitation (ChIP) and a combination of ChIP and the Self-Administered Gerocognitive Exam (SAGE), scientists have found many CREB sites that are linked to the neoplastic phenotype, clonogenic potential, resistance to apoptosis, and other features of abnormal growth [115,116,117,118]. Moreover, transgenic mice that overexpress CREB develop myeloproliferative diseases [106]. Furthermore, CREB has been connected to the development of resistance to inhibitors of the Raf-MEK-ERK and PI3K/AKT signaling pathways [119,120]. In breast cancer, CREB may make it harder for MAPK inhibitors to work, which is also linked to changes in histone acetylation [119,121]. Additionally, when CREB is turned down, breast cancer 1 (BRAC1) expression changes and aromatase expression keep increasing. Aromatase is a key enzyme in the production of estrogen. Its transcription is controlled by CREB and is linked to the development of tamoxifen resistance [120].

Moreover, cAMP has been shown to inhibit Sirtuin 6 (SIRT6) expression and thereby diminish non-small-cell lung carcinoma (NSCLC) cell death caused by radiation [122]. Regulator of G protein signaling 17 (RGS17) stimulates cell proliferation through the cAMP–PKA–CREB pathway, which is elevated in 80% of lung cancer tissues relative to matched normal lung tissue [123]. The hypoxic response in lung cancer cells may be regulated via the cAMP–PKA–CREB pathway [124]. In contrast to the cAMP-Sirt6 pathway’s suppression of radiation-induced NSCLC cell death [122], the cAMP–PKA–CREB pathway seems to have an anticancer effect in radiotherapy. Forskolin pretreatment inhibited ataxia-telangiectasia mutated (ATM) and nuclear factor-κB (NF-κB through PKA-induced protein phosphatase 2 (PP2A) phosphorylation, resulting in an increase in radiotherapy-induced apoptosis [125]. Additionally, the cAMP–PKA–CREB pathway is involved in the metabolic control of breast cancer. In breast cancer cells, serotonin increases mitochondrial biosynthesis through the AC-PKA pathway [126]. The cytoplasmic G-protein-coupled estrogen receptor increases aerobic glycolysis through the cAMP–PKA–CREB pathway [127]. These findings suggest that the cAMP–PKA–CREB pathway may have varied effects on the same kind of tumor depending on the circumstances.

## 4. Involvement of EPAC in Tumor Growth

EPAC has two roles in controlling how cancer grows and spreads. Most studies have shown that EPAC makes cancer cells grow and spread, but others have shown that it keeps cancer cells from spreading [29,128,129,130,131]. The difference between these results could be due to the different types of cells that were studied or to changes in the genomes and transcriptomes of the cancer cell lines. In general, EPAC and PKA-mediated signaling pathways either antagonistically, independently, or synergistically influence cancer cell proliferation, apoptosis, adhesion, and migration [129,130,132,133,134,135,136]. In the context of cAMP/PKA signaling, it has long been implicated in cancer formation and progression. The correlations of EPAC proteins, notably EPAC1, with cancer are developing and have been reviewed in a study [137]. Activation of the cAMP/EPAC1 signaling pathways has also been shown to make resistant cancer cells more susceptible to oncolytic virotherapy [138,139]. EPAC1’s effects on cancer cell proliferation and survival are cell-type- and context-dependent. While EPAC1 reduces cell proliferation in A498 clear renal cell carcinoma (cRCC) cells [140], it promotes cell proliferation and survival in prostate cancer cells by upregulating Ras/MAPK and PI3K/Akt/mTOR signaling [141,142,143]. Similarly, EPAC1 expression is enhanced in human ovarian cancer cells, and silencing EPAC1 with short interfering RNA decreases proliferation and promotes cell cycle arrest in vitro as well as suppressing tumor development in vivo in xenograft nude mouse models. Downregulation of EPAC1 in ovarian cancer cells greatly reduces phosphorylated protein kinase B (pAkt), cyclin D1, and cyclin-dependent kinase 4 (CDK4) signaling [144]. Research implicates the EPAC1/Rap1 signaling pathway in promoting oncogenesis by upregulating aerobic glycolysis [145]. EPAC1 may possibly give growth and survival benefits to cancer cells via metabolic reprograming. Several malignancies, including melanoma [146,147], prostate cancer [148], ovarian cancer [149], pancreatic cancer [150,151], cervical cancer [152], fibrosarcoma [153], and lung cancer [154,155], have been associated with EPAC1 invasion and metastasis.

Most studies show that activating EPAC1 makes cancer cells move and spread, but a few studies that used EPAC-selective agonist 007 show that activating EPAC1 stops cancer cells from moving [156]. However, some researchers claim that the inhibitory effect seen for EPAC1 in the contradictory studies was caused by the indirect activation of PKA. This is because the inhibitory effects caused by the 007 inhibitor could be reversed by PKA inhibitors H89 and protein kinase inhibitor peptide (PKI) but were unaffected by silencing the expression of EPAC1 and EPAC2 with siRNA [148]. Metastatic melanoma has higher levels of EPAC1 expression than primary melanoma, and EPAC1 expression is linked to the expression of N-deacetylase/N-sulfotransferase-1 (NDST-1) and heparan sulfate (HS), which is a major part of the extracellular matrix [147]. Moreover, the production of HS in response to more NDST-1 is linked to the cell migration caused by EPAC. Moreover, the movement of syndecan-2 (Sdc2), an HS proteoglycan on the surface of cells, to lipid rafts is controlled by EPAC1/PI3K-dependent tubulin polymerization [157]. The movement of melanoma cancer cells caused by EPAC1 has also been linked to PLC/IP3 receptor-dependent intracellular Ca^2+^ signaling and actin assembly [158]. Similarly, EPAC1 is highly expressed in pancreatic cancer [159]. Genetic and pharmacological studies show that EPAC1 helps pancreatic cancer cells move and spread by making integrin 1 become active and move around [150]. Studies also show that EPAC signaling can help cancer cells move by increasing the expression of histone deacetylase 6 (HDAC6) [155] or by making it easier for β-catenin to move into the nucleus and increase transcription [154] in lung cancer cells. Even though there are many different signaling pathways involved in EPAC1-driven cancer cell migration and invasion, most of them may eventually converge and link this EPAC1 function to a mechanism that depends on integrins [150,158]. Most of the evidence that links EPAC to cancer provided evidence that used cancer cell lines or xenografts to grow tumors in animal models. A study of 141 people with gastric cancer showed that EPAC1 expression is higher in the cells and tissues of individuals with gastric cancer. Importantly, the overexpression of EPAC1 is linked to several clinicopathological parameters, such as the depth of invasion, the stage of the cancer, and the spread of the cancer to the blood vessels. A Kaplan–Meier analysis also shows that the upregulation of EPAC1 is significantly linked to both poorer overall survival and disease-free survival, which suggests that EPAC1 can be used as a prognostic marker to predict gastric cancer [160]. In the same way, positive EPAC1 expression was found in 63 percent (32/51) of invasive ductal esophagus cancer tissue samples, which was a lot more than the 20 percent (2/10) positive expression rate found in para-carcinoma tissue samples [161]. An analysis of the Cancer Genome Atlas (TCGA) dataset shows that EPAC1, PKA, A-kinase anchoring protein 9 (AKAP9), and other cAMP signaling components are amplified in breast cancer patients, and this is linked to a lower chance of survival. Moreover, the pharmacological inhibition of EPAC1 by an ESI-09 inhibitor stops breast cancer cells from growing and moving, stops the cell cycle, and causes them to die [162]. The expression of EPAC1 was found to be higher in 58 percent (29/50) of breast cancer patients compared with a 10% positive rate (1/10) in controls [163]. These studies show that EPAC1 could be used in new and exciting ways to treat cancers of the breast, stomach, and esophagus. So far, almost all the links between EPAC signaling and cancer have been made with EPAC1. A study found that cAMP increased the expression of histone deacetylase 8 (HDAC8) and increased the apoptosis caused by cisplatin in H1299 lung cancer cells in a way that was neither PKA-dependent nor EPAC-dependent. Surprisingly, cAMP’s effect was caused by EPAC2, not EPAC1, because silencing EPAC1 with an EPAC1-specific shRNA did not stop the increase in HDAC8 expression caused by cAMP. However, EPAC2 shRNA or an EPAC2-specific inhibitor, ESI-05, did stop the increase in HDAC8 expression caused by cAMP. The authors think that turning on EPAC2 slows down the PI3K/Akt/MKK4/JNK1 pathway, which in turn, slows down the breakdown of the HDAC8 protein.

Furthermore, mTOR is a serine/threonine kinase that controls cell growth [164]. This speeds up the death of lung cancer cells caused by cisplatin by stopping the expression of the mTOR signaling pathway regulator-like (TIPRL) protein [165]. Interestingly, increasing scientific evidence shows that EPAC causes prostate cancer cells to grow (Figure 4). EPAC turns on the extracellular-signal-related kinase 1/2 (ERK1/2) and PI3K/Akt signaling pathways, which are both connected to mTOR signaling [166]. It might also increase the production of chronic inflammatory markers such as cytosolic phospholipase A2 (c-PLA2), cyclooxygenase-2 (COX-2), and prostaglandin E2 (PGE2). One theory says that when EPAC stimulates cells, COX-2 converts arachidonic acid, which is made when c-PLA2 activity goes up, into PGE2. When PGE2 binds to its receptors, prostaglandin E2 receptor 2 (EP2) and prostaglandin E2 receptor 4 (EP4), the cAMP/EPAC/Rap1 pathway is started, which causes mTOR signaling [167]. EPAC also raises the levels of cell cycle regulators such as cyclin B1 and cyclin-dependent kinase 1 (CDK1), which allows cells to move from the G2 phase to the M phase and speeds up the process of mitogenesis [142].

In addition to its contrasting effects in distinct cancer types, EPAC also exhibits opposing effects in cancer cell lines. For example, EPAC, like PKA, has contradictory effects on cell growth in neuroendocrine tumors (NETs). To promote pancreatic NET cell proliferation, EPAC enhances the level of cyclin D1, while lowering that of p27, a CDK inhibitor [130]. Raf-1 proto-oncogene, serine/threonine kinase (Raf1) is prevalent in pancreatic NETs and bronchial carcinoids. cAMP promotes MAPK and consequently cell proliferation via B-Raf while inhibiting MAPK via Raf1 [168]. Similarly, EPAC has pro-cancerous effects in patients with blood malignancies. Unlike PKA, EPAC enhances B-cell chronic lymphocytic leukemia (B-CLL) survival by activating Rap1 (Figure 5A) [169]. EPAC also inhibits the pro-apoptotic activity of PKA in acute lymphoblastic leukemia (ALL) cells [129] by first activating Rap1 and H-Ras, which in turn, boosts the stimulation of ERK1/2 and Akt (Figure 5B) [128]. In conclusion, more study is clearly required to determine whether the dysregulation of EPAC2 is linked in cancer like that of EPAC1.

## 5. cAMP and Its Other Effectors Act in Various Signaling Pathways

The MAPK pathway driver mutations are found in most cutaneous melanomas [170]. Proliferative signals from the cell’s surface RTK are sent progressively via the RAS, RAF, MEK, and ERK proteins in the conventional MAPK pathway [171]. c-KIT is an RTK present in melanocytes that, when bound to the stem cell factor (SCF), activates the small GTPase RAS [172]. It is the RAS kinase that then activates the RAF kinase (MAP3K) family. In turn, mitogen-activated protein kinase (MAP2K) and ERK are triggered by the RAF kinase family, which includes B-Raf proto-oncogene (BRAF) and C-Raf proto-oncogene serine/threonine protein kinase (CRAF) [173]. The phosphorylation and activation of ERK by microphthalmia-associated transcription factor (MITF) and other downstream targets in melanocytes helps regulate several cellular processes. The MAPK pathway and cAMP signaling have been shown to have several layers of interaction. The cAMP signaling pathway that inhibits CRAF is reliant on PKA [174], while the pathway that activates BRAF is dependent on EPAC [175]. PDZ-GEF1, also known as RapGEF2 or CNrasGEF, is a RAS guanine nucleotide exchange factor with a cAMP/cGMP-binding domain that may activate RAS via cAMP signaling [174,176]. Furthermore, CREB is believed to promote the production of the RKIP, a well-known inhibitor of RAF kinase [177]. Cancers, including melanoma, are known to have decreased levels of RAF kinase inhibitory protein, which results in increased activation of downstream MAPK signaling [178,179]. The activation of the MAPK pathway by BRAF and RAS gene mutations is known to increase melanoma growth, invasion, metastases, and angiogenesis [180]. As cAMP signaling and the MAPK pathway cross speak, it is plausible that cAMP signaling may play a role in cancer development. cAMP’s significance in melanoma is unclear, and research looking at how cAMP signaling impacts melanoma typically provides conflicting findings.

Moreover, the Popeye domain containing (POPDC) protein family is one of the five downstream targets of cAMP. The other four are PKA, EPAC, hyperpolarization-activated cyclic nucleotide-gated (HCN), and cyclic nucleotide receptor involved in sperm function (CRIS) [181]. cAMP binds to the phosphate binding cassette (PBC) of the POPEYE domain, which causes more proteins to be made, stabilized, and turned on [182]. Higher levels of cAMP in cancer cells are linked to more apoptosis and less growth, invasion, and spreading of the cancer. Since PKA and EPAC, which are also downstream effectors of cAMP, make these effects stronger, this suggests that the pro-apoptotic effects of boosted cAMP in cancer cells could be caused, at least in part, by the actions of POPDC1 proteins [182]. POPDC1 also affects other signaling pathways, such as the Wnt pathway, and transcription factors, such as c-Myc. It interacts with proteins such as caveolin-3 (CAV-3), TREK1 (two-pore domain potassium channel), TJ-associated proteins (such as ZO-1 and occludin), guanine nucleotide exchange factors (GEFT and GEFH) [183,184], and the vesicular transport protein VAMP3 [185]. So far, the interactions of POPDC1 with TREK1, CAV-3, B-cell lymphoma-2 (Bcl-2), and adenovirus E1B 19 kDa interacting protein 3 (Bnip3) have mostly been shown in cardiac and skeletal muscle function [186]. However, recent studies have found that TREK1 is overexpressed in prostate cancer (PC) cells, CAV-3 is upregulated in anaplastic thyroid carcinoma [185], and Bnip3 levels are higher in breast cancer (BC) and NSCLC [187]. However, it has not been proven that POPDC1 is in cancer cells or that it interacts with these proteins. The known effects of POPDC1 on cancer cell targets will be discussed briefly in the following paragraph.

POPDC1 proteins have been shown to be downregulated in a variety of cancer cell types. Four primary pathways have been postulated to cause the loss of POPDC1 expression. First, the hypermethylation of the POPDC1 gene promoter’s cytosine–phosphate–guanine islands has been seen in a variety of tumors (Table 1) and has been proposed as a biomarker for early cancer detection [183]. Second, POPDC1 expression is known to be suppressed by an underexpression of microRNA (miRNA)-122 [188] and, thirdly, an overexpression of netrin-1 [189]. Finally, increasing EGFR stimulation causes POPDC1 activity to be significantly suppressed, most likely due to the phosphorylation of the POPDC1 c-terminal domain (CTD) [190]. Table 1 lists the pathways implicated in various malignancies. POPDC1 is no longer able to impact cell adhesion and interact with the numerous signaling pathways and proteins listed above (Table 1) due to its lack of expression and membrane integration.

Furthermore, somatic mutations in the PDE4DIP gene have been identified in individuals with drug-resistant prostate cancer [198], endometriosis-associated ovarian cancer [199], and familial squamous lung cancer cancers [200]. Mutations in the phosphodiesterase 4D-interacting protein (PDE4DIP) gene have also been found in NSCLC with leptomeningeal metastases [201]. Germline indels and single nucleotide variations in the PDE4DIP gene have been found in leukemia patients [202]. This PDE4DIP function is necessary for cell stability at the leading edge of migrating cells, and its influence on tumor cell motility and proliferation is vital in tumor development [203]. PDE4DIP deficiency has also been demonstrated in a mouse model to limit the proliferation of granule neuron precursors and to inhibit the formation of medulloblastoma [204], highlighting its significance as an appealing target for the treatment of malignant diseases. Myomegalin antibodies have been found in the serum of patients with esophageal squamous cell carcinoma (SCC) and are linked to a better prognosis [205]. PDE4DIP may be found in up to three copies on chromosome 1q21 [206]. Interestingly, pineoblastoma cells carry up to eight copies of the gene [207]. It is worth noting that the PDE4DIP gene has a single DUF (domain of unknown function), the 1220 domain (DuF1220) [208]. The DuF1220 domain has the highest copy number increase in the human genome, and its copy number has been linked to abnormal brain sizes (same reference). Overall, our data indicate that PDE4DIP may play a role in tumor development by promoting tumor cell proliferation and migration, making it an appealing target for the treatment of a variety of malignant tumors.

## 6. Potential Anticancer Therapeutic Strategies

Some commercial drugs and peptides (such as forskolin, CREBtide, and KEMPtide) as well as chemical treatments (such as zinc sulfate) have been shown to interfere with the production or breakdown of cAMP as well as increase or decrease PKA activity [209]. The eight-substituted and six-substituted forms of cAMP are the ones that PKA type I and type II regulatory subunits like to bind to. With the goal of selectively increasing PKA RI, these cAMP analogs have been looked at as possible anticancer drugs [210,211,212]. Targeting the catalytic subunit of PKA may disrupt its function since it is believed to serve as a scaffold for a variety of interactions with other proteins [17]. Additional anticancer medications, such as phosphodiesterase inhibitors, have also been utilized in tandem with traditional chemotherapy [213,214]. Targeting PKA has been studied as a way to treat lung cancer because it is involved in acetylcholine receptor signaling [215]. In a rat model of acute myeloid leukemia, activating type II PKA showed anti-leukemic effects [216]. As recently shown in several cancer cell lines, accessory proteins such as AKIP1 control the PKA-induced activation of NF-κB, allowing the precise prediction of the impact of PKA inhibition as a cancer treatment [217]. Kinase activation is one of the components of kinase function modulation. When the transcription of one PKA regulatory isoform is repressed, expression of another isoform is increased as a compensatory effect [209]. The anticancer effect of PKA antisense oligonucleotides [218,219] was explored to suppress Riα expression in malignancies, where RIα appears to play a key role.

CREB is an ideal target for the treatment of malignancies because of its involvement in tumor growth, maintenance, and progression. The expression of CREB is decreased in the bone marrow cells of individuals with acute myeloid leukemia (AML). For example, CREB is involved in several signal transduction pathways linked to tumor growth. CREB function in tumor cells can now be inhibited using several approaches (Figure 6). The use of dominant-negative CREB mutants (KCREB) may limit CREB transcription by heterodimerizing KCREB with wild-type CREB, among other approaches. In vitro and in vivo studies showed that KCREB overexpression in metastatic tumor cells reduces the likelihood of metastasis [220]. By combining the dominant-negative A-CREB with photoactive yellow protein, researchers discovered a novel form of CREB inhibitor [221]. A link between CREB and the optogenetic domains made these discoveries possible. This link allows studies investigating CREB’s ability to control space and time and its potential as a therapy. Many “decoy” oligonucleotides have been made that stop CREB gene transcription and slow tumor growth [222]. RNA interference inhibits CREB expression, causing alterations in cell growth and survival. Inhibiting CREB in tumor cells lowers tumor development in vivo, inhibits cancer cell proliferation, migration, and anchorage-independent growth, suppresses cell cycle arrest, promotes apoptosis, and enhances tumor immunogenicity [104,223]. CREB is also affected by several pathways that have been linked to the growth of tumors. As a result, small-compound CREB inhibitors have been developed as “proof of concept”, and investigations have shown CREB inhibition’s therapeutic potential. Chemical inhibitors of the CREB-cAMP response element (CRE) or CREB-CREB binding protein (CBP) interaction as well as other kinase inhibitors that prevent CREB phosphorylation and activation have been discovered [224,225,226,227,228]. The KID-KIX complex interaction between the CBP and CREB proteins may be slowed down by the inhibitor KG-501 in a dose-dependent manner, which can clearly suppress transcription, even at micromolar quantities. As an alternative, microRNA could be used to stop the expression and activity of CREB microRNA that directly stop CREB activity [113] and stop tumor cells from becoming cancerous, but their use in living things has not yet been proven.

Furthermore, EPAC has been linked to malignant cell proliferation, migration, metastasis, and death. Thus, EPAC has been widely studied as a potential therapeutic target for cancer. Several in vitro and in vivo investigations have shown that EPAC modulation is a viable therapeutic option for cancer. The therapeutic value of EPAC inhibitors [160,229] and activators [230] varies depending on cancer type. EPAC modulators may be utilized as standalone chemotherapeutic medicines or adjuncts in cancer therapy strategies. Furthermore, EPAC suppresses both regulatory T cells (Tregs and Teffs) [231]. These side effects also hurt T-cell-based cancer immunotherapies, making them less effective at killing tumors [232]. The combination of immunotherapy and EPAC inhibitors may protect injected T cells from EPAC-mediated inhibition and hence retain their therapeutic effects. Studies have shown that the EPAC inhibitor alone (ESI-09) or lithium [233,234,235] exerts a considerable inhibitory effect. EPAC inhibitors may therefore operate in tandem with other chemotherapy medicines. On the other hand, EPAC activators may boost the effects of ionizing radiation and chemotherapeutic drugs such as topoisomerase II inhibitors. In malignant cells, radiotherapy and many chemotherapeutic treatments have been shown to cause double-stranded DNA breaks. DNA-dependent protein kinase (DNA-PK) aids cells in repairing and rescuing themselves from DNA damage [236]. DNA-PK inhibitors have been proven in vivo and in vitro to sensitize tumor cells to chemo and radiation [237,238]. Interestingly, EPAC helps DNA-PK move into the nucleus, separates it from its substrates, and stops double-strand break repair [239]. In this context, the therapeutic significance of numerous EPAC inhibitors and activators is being investigated. EPAC inhibitors have been demonstrated to inhibit one of the two EPAC isoforms selectively [240]. For instance, ESI-09 has a competitive antagonistic impact on EPAC1. Alternatively, ESI-09’s effects might be linked to its wide protein-denaturation characteristics [240]. Another study [235] corroborated ESI-09’s selectivity and found that its protein-destabilizing effects were not significant at pharmacologically effective doses. ESI-09 has shown great bioavailability and safety in animal experiments [241]. CE3F4R, another EPAC1-selective inhibitor, binds to the EPAC1-cAMP complex non-competitively [33,242]. Conversely, the EPAC2-selective drugs ESI-05 and ESI-07 work on a recently identified allosteric site in EPAC2 that links two CNBDs; this effect is not seen in EPAC1 since it only has one CNBD. When ESI-05 or ESI-07 binds to this domain, it stops EPAC2 from becoming active [33,242]. Compounds activate EPAC such that one of the two isoforms is activated. cAMP analogs that cannot activate other cAMP-dependent proteins, particularly PKA, are among these activators. EPAC1 is activated by 8-pCPT-2′-O-Me-cAMP, a selective EPAC activator [243]. 8-pCPT-2′-O-Me-cAMP-AM is a potent prodrug that is hydrolyzed to the active form after crossing the cell membrane [244,245]. In addition to EPAC1-specific activators, Sp-8-Bnt-Me-cAMPS is a cAMP analog that activates EPAC2 while inhibiting EPAC1 [33,246]. Sulfonylureas, which are already approved for usage in diabetic patients, have been demonstrated to activate EPAC selectively [247]. They represent a promising pool of potential cancer targets as well as inhibitors of critical processes that feed cancer survival and development.

## 7. Updated Potential Anticancer Therapeutic Strategies

As tumor growth is linked to the cAMP–PKA system, targeting this pathway may be an effective cancer treatment strategy. The cAMP analog 8-CL-cAMP, more commonly known as tocladesine, has been shown in vitro and in vivo to prevent the development of many cancers, including breast, lung, fibrosarcoma, and leukemia [248,249]. 8-Cl-cAMP, which has a lower affinity for PKA R subunits than RI, may suppress RI expression while increasing RII expression [248]. According to some studies, 8-Cl-metabolite cAMP’s 8-Cl-adenosine and the AKT2–PKB pathway may also have anti-tumor effects that are PKA-independent [250,251]. Even though it is not clear how 8-Cl-cAMP kills tumors, there have been published phase II clinical studies looking at its effectiveness in treating multiple myeloma and phase I clinical studies looking at how it works in treating metastatic colorectal cancer (Table 2, data retrieved from www.clinicaltrials.gov (accessed on 15 June 2022)). Some types of cancer may be inhibited by increasing cAMP levels due to its tumor-suppressive properties. Preventing cAMP depletion may be achieved using PDE inhibitors. Furthermore, PDE inhibitors are often utilized for treating cardiovascular, pulmonary, and psychiatric conditions in the clinical setting. Amrinone and milrinone, the two PDE3 inhibitors, are cardiotonic drugs. Rolipram, a novel anti-inflammatory drug for treating asthma, inhibits PDE4 in asthma patients. Sildenafil, a PDE5 inhibitor, is the preferred therapy for erectile dysfunction [252]. PDEs may also be inhibited by certain flavonoids found in natural products [253]. Repurposing PDE inhibitors for the treatment of malignancies wherein cAMP shows tumor-suppressive effects may be an option in the future. In contrast, cancers that are fueled by cAMP signaling could benefit from PDE activators. An allosteric activator of the PDE4 long isoform was recently discovered [254]. In the kidneys, this prototype PDE4 activator inhibits cyst formation by reducing intracellular cAMP levels [254]. However, whether these PDE activators may be used to treat some forms of cancer remains unclear. Another option for interfering with PKA signaling is using antisense oligonucleotides targeting N-terminal PKA RI. As a result of PKA RI downregulation, tumor development is inhibited in a broad range of tumor models [255]. Two phase I clinical studies examined the antisense oligonucleotide GEM231 (Table 2, data retrieved from www.clinicaltrials.gov (accessed on 15 June 2022)). On average, following 39 cycles of GEM231 plus docetaxel treatment, 75 percent of 20 patients with resistant solid tumors had grade 3 side effects such as fatigue, elevated aminotransferase activity, neutropenia, and altered sensorium [256]. This clinical study has not been previously mentioned.

In addition, CREB is a candidate therapeutic target for cancer, although at present no CREB inhibitor is available commercially. CBP is a coactivator of CREB, and the small chemical XX-650-23 disrupts their association, resulting in apoptosis and cell-cycle arrest in AML cells [257,258]. Downregulation of CREB expression is another therapeutic option. GSKJ4, an inhibitor of the histone lysine demethylases Jumonji domain-containing protein-3 (JMJD3) and ubiquitously transcribed X chromosome tetratricopeptide repeat protein (UTX), causes CREB degradation and suppresses the proliferation of AML cells [259]. In addition to CREB, GSKJ4 inhibitor targets may include JMJD3/UTX, which in turn has several targets. To suppress CREB, blocking calcium/calmodulin-dependent protein kinases (CAMK), which activate CREB in certain cancers, is another potential strategy [260]. The tumor-suppressing effects of several PKA inhibitors have been observed in preclinical investigations; however, no small-molecule PKA inhibitors have yet been evaluated in human clinical trials. Research on the development of new small-molecule medicines targeting PKA and/or CREB in cancer treatment is needed. In addition, cancer treatment may benefit from the repurposing of several PKA- and CREB-targeting medicines. Patients with cancer may benefit from the use of these tumor-targeted therapies.

The fact that EPAC modulators are believed to be safe and to have few adverse effects makes them even more attractive for use in clinical settings. Since EPAC has distinct isoforms, each may be addressed separately [261]. Therefore, it is reasonable to assume that blocking or activating one of them will have tissue-type effects with minimal side effects. EPAC is also abundantly expressed in cancer cells and is reliant on cell proliferation. This has been observed in cell culture as well as in patient-based studies [262,263]. A cohort-based investigation reported significantly higher levels of EPAC in stomach cancer cells than in other tissues [160]. EPAC overexpression has been observed in tumor cells relative to non-malignant cells [163]. Modifying EPAC activity may thus have a more powerful impact on cancer cells. Interestingly, EPAC-deficient mice do not show any signs of malnutrition, providing evidence that altering EPAC expression has no detrimental effects on developmental processes [262]. Although EPAC-targeting medications have not been adequately evaluated in cancer, their adverse effects are negligible when used for treating cardiovascular illnesses. Moreover, EPAC modulators are less likely to cause cardiac failure than beta-blockers [264]. These medications may also be used as alternatives to opioids for pain control. Furthermore, in vitro and in vivo testing is needed to assess the chemotherapeutic potential of these medicines in clinical settings.

Various studies have examined the role of cAMP in melanoma responsiveness to therapy. More than 15,500 genes were specifically expressed in a BRAF V600E melanoma cell line to assess the influence of each gene on the susceptibility of melanoma to MAPK pathway inhibition in a systematic gain-of-function resistance study [119]. The overexpression of genes encoding important components of plasma membrane cAMP signaling microdomains, such as GPCRs, tmACs, PKA, and CREB, was shown to be related to resistance in this study [119]. Furthermore, a higher expression of adipocyte enhancer-binding protein 1 (AEBP1) has been shown to confer resistance to BRAF inhibition in vivo, and greater CREB activation is required for AEBP1 overexpression in resistant melanoma cells [265]. These findings indicate that cAMP signaling may play a role in treatment resistance.

In contrast, another study revealed that suppressed cAMP signaling is linked with resistance to BRAF inhibitor treatment. Resistance to BRAF inhibitors is related to reduced cAMP levels in *BRAF* wild-type and *NRAS* wild-type melanoma cells and restoring cAMP levels using forskolin (FSK), a cAMP activator, while IBMX, a universal inhibitor of PDEs, sensitizes melanoma cells to BRAF inhibitors [266]. Furthermore, recent research has shown that melanoma cells are more susceptible to immunotherapy when cAMP signaling downstream of the G-protein-coupled estrogen receptor (GPER) is activated. The combination of G-1 pretreatment and the systemic delivery of an anti-programmed cell death 1 (PD-1) antibody inhibited tumor development and extended the life of melanoma cells [267].

Several studies have investigated cAMP signaling using pharmacological drugs, such as FSK, which promotes tmAC activity by binding to its allosteric regulatory region [268]. In contrast, FSK has no effect on sACs [269,270]. Nevertheless, the function of sAC in cAMP-dependent actions in melanoma remains to be fully investigated. Because various intracellular cAMP microdomains contribute to different cellular processes, studying cAMP at the microdomain level will help to better understand why this second messenger has such a diverse impact on melanoma biology.

A recent study found that Ca^2+^ controls cAMP production either directly or through calmodulin [271]. This study was interested in analyzing the regulatory effect of PRP4 on AC and cAMP in relation to the effect of PRP4 overexpression on intracellular Ca^2+^. Reducing the expression of PRP4 by using siRNA dramatically restored the expression of AC [272]. Finally, the effects of PRP4 overexpression on cAMP production were evaluated. PRP4 overexpression also decreased cAMP production, which was reversed when an siRNA-induced PRP4 knockdown was used [272]. Studies have shown that cAMP controls RhoA and alters the morphology of cells [273]. PRP4 overexpression has previously been shown to alter cell morphology from a flattened, aggregated form to a spherical shape by inhibiting RhoA activity [274]. A change in the B16F10 cell actin cytoskeleton shape is caused by the regulation of RhoA by PRP4. As a result, the PRP4 controlled the B16F10 cell actin cytoskeleton without interfering with nuclei [272]. The AC–cAMP–RhoA pathway seems to be the mechanism through which PRP4 alters the morphology of B16F10 cells. The usage of these tumor-specific medicines may be beneficial for cancer patients.

## 8. Conclusions

cAMP signaling in cancer cells is affected by the type of cell and its surroundings. Human malignancies are linked to the cAMP–PKA–CREB signaling system. When this system is turned on and linked to other signaling pathways, it can lead to the growth of tumors. In Hodgkin’s disease, cAMP and CREB are tumor suppressors [275,276,277]. In mice, CREB activity is deadly when it is disrupted. Genetic models may be useful in the development of cellular survival and maintenance as well as their participation in illnesses. Reversing CREB-induced cellular transformation will be possible if the molecular mechanisms that control CREB expression are identified. Cancers with active signal transduction pathways may have CREB as a prognostic and therapeutic target. PKA also blocks signals from cyclin A, ERK, and other proteins that cause tumors. PKA phosphorylates Raptor on Ser791 to inhibit mTORC1 activity [278,279]. If cAMP and its effectors may treat cancer and overcome medication resistance, more research is needed. To help in the development of new cancer therapy options, it is necessary to discover the downstream effectors of EPAC signaling that mediate its contradictory effects in diverse cancer types and the cell lines generated from those cancers. Replications of the in vitro and vivo results need further investigation, especially in clinical research. Furthermore, efforts must be made to better understand the cAMP signaling pathway and its associated effectors. Recent advances in high-throughput and high-content screening methods may be used to accelerate the search for inhibitors and agonists of the cAMP signaling system. Learning more about the significance of the cAMP signaling pathway in pathologies, particularly cancer, where signals fail or are disrupted, might aid in the construction of a therapy strategy based on stimulating the pathway. Advances in research, such as the involvement of PRP4 in cAMP signaling [272], may pave the way for new techniques to understand and develop cancer therapies.

## Figures and Tables

**Figure 1 cells-11-02020-f001:**
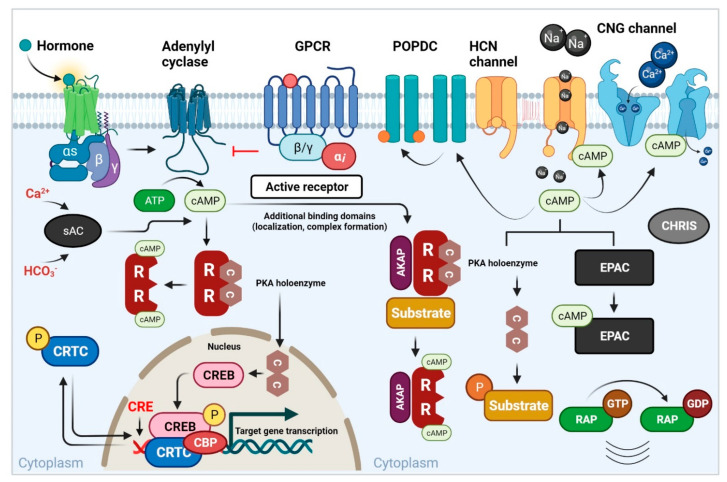
**An overview of mammalian second messenger cAMP signaling pathways** [10]. Upstream stimulation of Gαs-coupled GPCRs, which then activate AC to create cAMP, is required for signaling through the PKA pathway. The activation of Gαi-coupled GPCRs inhibits the synthesis of AC and cAMP. Ca^2+^ and HCO_3_^−^ activate soluble AC (sAC), which leads to cAMP synthesis. The generation of cAMP in the cell is regulated by multiple ACs as well as its breakdown by PDEs. The tetrameric PKA holoenzyme is made up of two R subunits and two C subunits. Regulatory subunits and substrates are coordinated by AKAPs. Additional binding domains on AKAPs aid in the building of protein complexes and allow them to be targeted to specific sites inside the cell. When cAMP binds to regulatory subunits, the holoenzyme dissociates, allowing catalytic subunits to phosphorylate substrates. CREB-mediated transcription is mediated by PKA. A hormone binds to Gαs-linked GPCRs on the cell surface, stimulating cAMP synthesis and PKA activation through adenylyl cyclase signaling. Adenylyl cyclase and cAMP generation are inhibited when Gαi-coupled GPCRs are activated. C subunits translocate to the nucleus to phosphorylate CREB on serine 133 when they are active. To enhance binding to CREs and transcription of target genes, phosphorylated CREB binds coactivators such as CBP. CREB-mediated transcription is regulated by other coactivators such as CRTCs. Phosphorylation of CRTCs by other kinases causes them to be sequestered in the cytoplasm, while dephosphorylation by phosphatase allows them to be translocated to the nucleus. Beyond PKA, cAMP binds to and activates effectors. cAMP modulates channel opening and cation currents through binding to CNG ion channels. HCN channels bind cAMP to help membrane hyperpolarization open the channel. In the Ras-associated protein (RAP) family of small GTPases, cAMP binds to EPAC to enable the exchange of GDP for GTP. POPDC proteins exist as dimers on the cell surface that bind cAMP. GPCR, G protein–coupled receptor; AC, adenylyl cyclase; PKA, protein kinase A; PDE, phosphodiesterase; AKAP, A-kinase anchoring protein; CREB, cAMP responsive element-binding protein; CRTC, cAMP-regulated transcriptional coactivator; CBP, CREB-binding protein; CRE, cAMP response element; CNG, cyclic nucleotide–gated; HCN, hyperpolarization-activated; POPDC, Popeye domain containing; EPAC, exchange protein directly activated by cAMP.

**Figure 2 cells-11-02020-f002:**
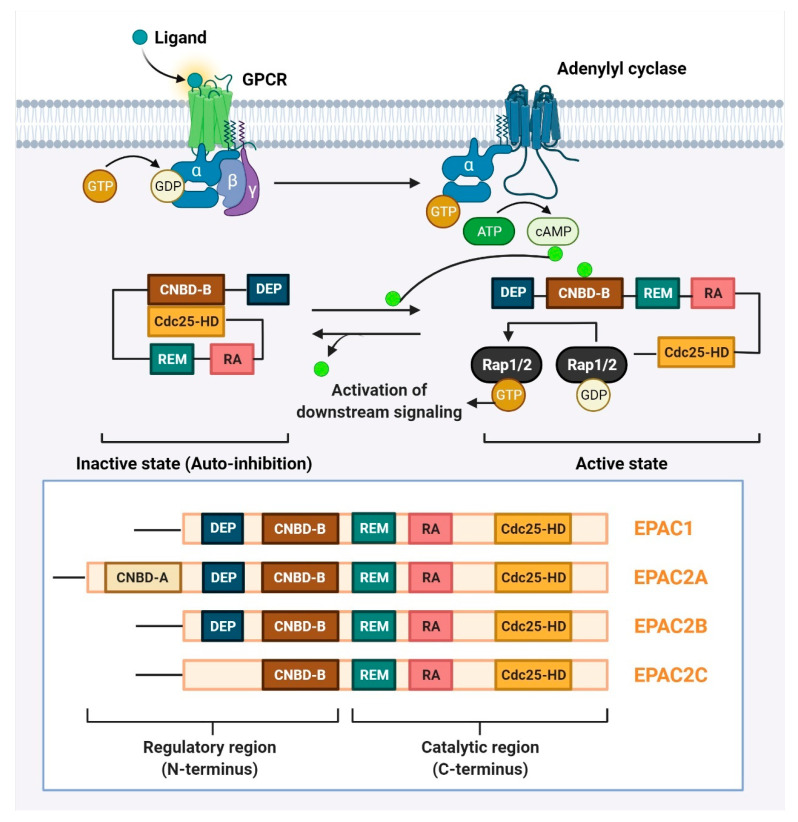
**Structure and mechanism of EPAC protein activation**.

**Figure 3 cells-11-02020-f003:**
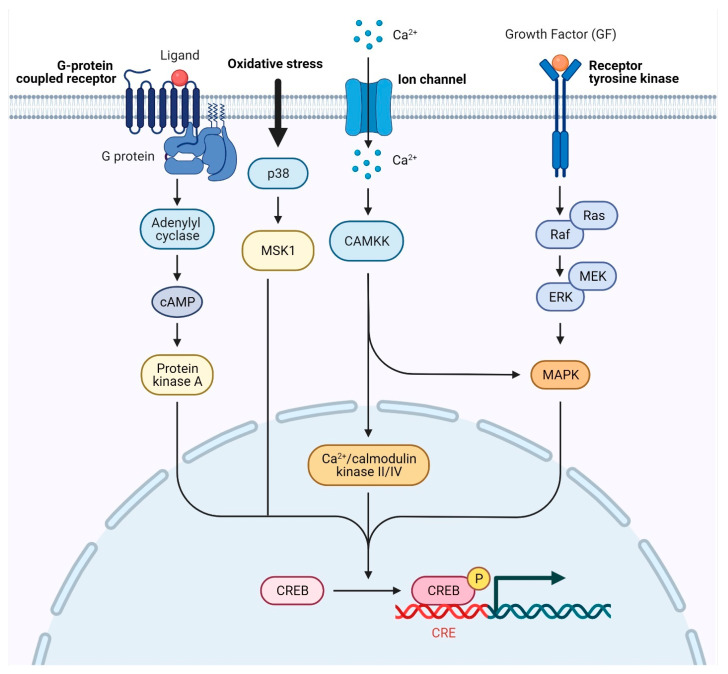
**CREB expression is regulated via signal transduction pathways** Cellular growth factors (GFs) may activate PI3K/AKT or Ras/MEK/ERK pathways when they connect to the membrane-bound receptor. Activation of calcium-dependent kinases increases as Ca^2+^ inflow increases. PKA is turned on when hormone receptors and G-protein-coupled receptors activate adenylate cyclase. All signal transduction pathways may phosphorylate CREB at different serine sites. PKA, protein kinase A; cAMP response element-binding protein, CREB.

**Figure 4 cells-11-02020-f004:**
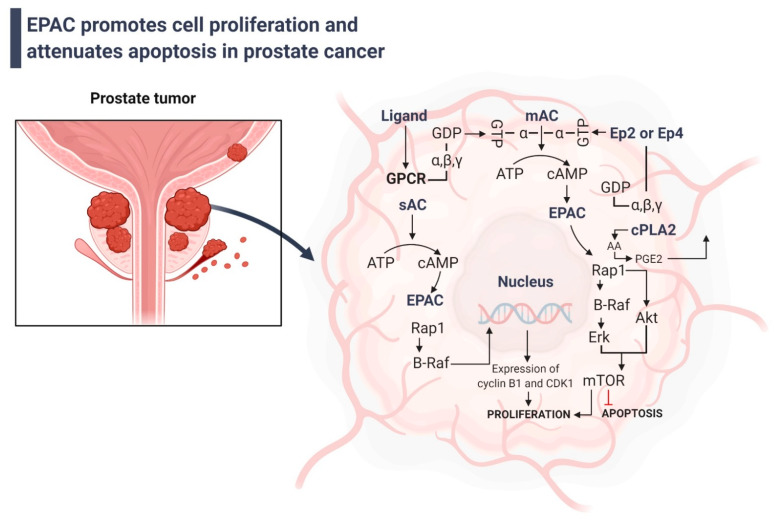
**EPAC stimulates cell growth and inhibits apoptosis in prostate cancer**. When sAC is turned on, it makes cAMP, which turns on EPAC, which is the soluble form of type 10 adenylyl cyclase. EPAC helps B-Raf, which then activates the expression of CDK1 and cyclin B1 in a way that depends on Rap1. These proteins help the cell cycle move from the G2 phase to the M phase. When mAC is turned on, EPAC is turned on, which causes cAMP to be made. The B-Raf/ERK and Akt pathways, which lead to mTOR, may then be turned on by EPAC. When mTOR is turned on, it helps cells grow and prevents them from dying. EPAC is anti-inflammatory, which makes these benefits even better. cPLA2 is turned on when MAPK is stimulated by EPAC. COX-2 changes phospholipids in the membrane into AA to make PGE2. PGE2 is made by prostate cancer cells and can move into the microenvironment of the tumor, where it can activate EP2 and EP4 receptors on target cells and cells close by. The mAC is also turned on by G proteins, which bind to the cAMP and EP4 receptors and cause these two molecules to build up. sAC, the soluble type 10 adenylyl cyclase; AA, arachidonic acid; COX-2, cyclooxygenase-2; cPLA2 cytosolic phospholipase A2; B-raf, Serine/threonine-protein kinase B-raf; EPAC, exchange protein directly activated by cAMP; EP2, PGE2 receptor 2; EP4, PGE2 receptor 4; mAC, membrane-bound adenylyl cyclase; PGE2, prostaglandin E2.

**Figure 5 cells-11-02020-f005:**
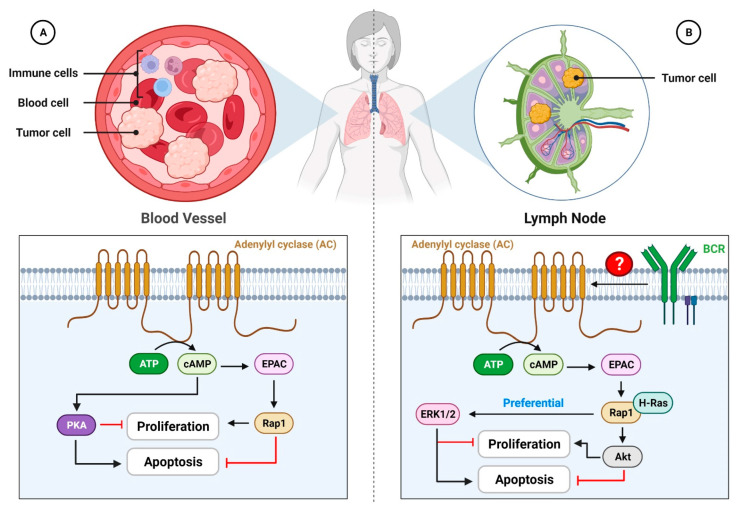
**Diverse blood cancers respond differently to EPAC.** (**A**) EPAC increases cell proliferation and survival while decreasing apoptosis in both B-cell chronic lymphocytic leukemia and acute lymphoblastic leukemia. Although both EPAC and PKA are downstream consequences of cAMP, their functions are diametrically opposed. EPAC, as opposed to PKA, inhibits apoptosis through Rap1 and promotes cell survival in the body. (**B**) EPAC induces cell growth arrest and death in immature B-cell lymphoma. To put it another way, cAMP levels rise when the BCR is activated. Accumulated cAMP then increases EPAC, which in turn, activates ERK1/2, which promotes apoptosis and Akt, which inhibits it, through Rap1 and H-Ras. Inhibition of cell growth and an increase in apoptosis are the likely outcomes of this activation, which seems to favor ERK. PKA, protein kinase A; EPAC, exchange protein directly activated by cAMP; BCR, B-cell antigen receptor; RAP1, Ras-related protein 1; ERK, extracellular-signal-related kinase.

**Figure 6 cells-11-02020-f006:**
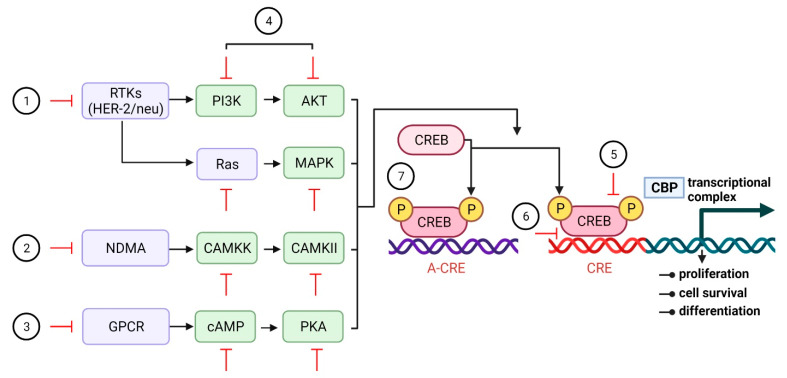
**Methods for suppressing CREB expression.** A variety of methods have been used in vitro and in vivo to reduce or abrogate CREB expression and activity: 1. Trastuzumab and lapatinib are upstream CREB inhibitors that block HER-2/neu and EGF-R (receptor tyrosine kinases), respectively. 2. Ketamine-induced suppression of ion transporters such as NMDA. 3. Use of beta blockers to inhibit G-protein-coupled receptors. 4. Distinct signal transduction inhibitors are used to examine kinase or substrate activity. 5. KG-501 prevents CREB from interacting with coactivators CBP/p300. 6. Utilizing surfen hydrate to influence CREB binding at the CRE site. 7. Employing a replica CRE element to limit CREB interaction with the gene promoter. Epidermal growth factor receptor, EGF-R; N-methyl-D-aspartate, NMDA; CREB, cAMP responsive element-binding protein; KG-501, 2-naphthol-AS-E-phosphate; CRE, cAMP response element.

**Table 1 cells-11-02020-t001:** cAMP/PKA functions and POPDC proteins’ roles, downstream targets, and downregulations associated with various cancer types.

Type of Cancer	cAMP/PKA Functions	Popeye Domain Containing Protein (POPDC) Cancer Types	Mechanisms and Roles of POPDC Proteins	POPDC Downstream Targets in Cancer Signaling Pathways and Protein Interactions
Squamous cell carcinoma ↑	Increasing the invasion and metastasis in the esophagus by PKA phosphorylating vasodilator-stimulated-phosphoprotein (VASP) [49].	POPDC1 in CRC, PC, BC, NSCLC, glioma, HNSCC, GC	Promoter hypermethylation [191,192,193,194]	POPDC1/ZO-1 protein interaction in trabecular meshwork cells, HCE, uveal melanoma prevents ZONAB-induced entry to cell cycle and translation of proliferative genes [195].
Lymphoblastic leukemia ↓	Autophagy, aided by cAMP-induced poly [ADP-ribose] polymerase 1 (PARP1) activation, may treat acute lymphoblastic leukemia [52].	POPDC1 in HCC	Underexpression of miRNA-122 [188] and overexpression of netrin-1 [189].	Occludin in HCE and uveal melanoma maintains tight junction formation [182,195].
Liver cancer	PKA phosphorylates many substrates, including CIP4, facilitating HCC invasion and metastasis [58].	POPDC2 in ductal breast carcinoma (especially HER2+ subtype)	Overexpressed at all clinical stages. Possibly implicated in cancer initiation and sustenance [190].	LRP6 (Wnt/βcatenin pathway) in HEK293 cells, human colonoids, murine adenoma tumoroids prevents β-catenin activation by inhibition of LRP6 [196].
	The vasoactive intestinal peptide lowered cAMP levels, CREB expression, and phospho-CREB (Ser133) phosphorylation via inhibiting B-cell lymphoma-extra-large (Bcl-Xl) expression [59].	POPDC3 in ductal breast carcinoma (especially HER2+ subtype)	Overexpressed at early clinical stages [190].	PR61α (c-Myc pathway) in murine colitis-associated cancer cells promotes c-Myc ubiquitination/ degradation [193].
	The catalytic subunit of PKA C (DNAJB1-protein kinase cAMP-activated catalytic subunit alpha (PRKACA)) was overexpressed, PKA activity increased [61].	POPDC3 in head and neck squamous cell carcinoma (HNSCC)	Overexpression correlates with low patient survival. Potential biomarker for radiotherapy resistance [197].	
Prostate cancer	The high PKA expression promotes cell proliferation and carcinogenesis [71].	POPDC3 in gastric cancer	Underexpression due to promoter hypermethylation. Lower POPDC3 levels correlate with increased depth of invasion and metastasis [192].	
	cAMP–PKA signaling pathway is required for high levels of osteocalcin and ostesialin production in androgen-independent prostate cancer [90].	POPDC3 in esophageal and lung cancer	Overexpression of POPDC3 correlates with greater radiotherapy resistance [197].	
	PKA activity may increase with depressive and behavioral stress [92,93].	LRP6 (Wnt/βcatenin pathway) interacting with POPDC1 in HEK293 cells, human colonoids, murine adenoma tumoroids	Prevention of β-catenin activation by inhibition of LRP6 [196].	
Small-cell lung cancer (SCLC) ↓	Inhibition of PKA activity [73].	Occludin interacting with POPDC1 in HCE, uveal melanoma	Maintenance of tight junction formation [182,195].	
Brain cancer	Stimulation of the cAMP pathway via PKA RII induces cell differentiation and death [74].			
	The catalytic subunit of PKA was found to be decreased in high-grade gliomas [76].			
	Increased cAMP levels reduce phosphatidylinositol 3-kinase, which decreases neuroblastoma [77].			
	Lower AC and cAMP levels in glioblastoma cells [79].			

**Note:** Promoter hypermethylation, underexpression of miRNA-122, which inhibits POPDC1 gene transcription, and overexpression of netrin-1, which phosphorylates and inactivates POPDC1, are the four primary mechanisms of PODPC1 downregulation. HCC, CRC, BC, PC, NSCLC, HNSCC, and glioma are only a few of the cancers that have been linked to these processes. Promoter hypermethylation is the most well-studied mechanism for POPDC1 downregulation. Many downstream proteins, such as ZO-1, occludin, LRP6, and PR61α, interact with the POPDC1 protein. This interaction has been shown primarily in cardiac and skeletal muscle cells. However, evidence suggesting POPDC1 interacts with these targets in cancer cells is accumulating. POPDC2 and POPDC3 expression vary depending on the type of cancer. POPDC2 dysregulation is mostly seen in heart disease and breast cancer. POPDC3 mutations have been linked to limb girdle muscular dystrophy and have been proven to have tumor-suppressive and oncogenic effects in various cancers. **Abbreviations:** VASP, vasodilator-stimulated-phosphoprotein; PARP1, cAMP-induced poly [ADP-ribose] polymerase 1; Bcl-Xl, B-cell lymphoma-extra-large; PRKACA, catalytic subunit of PKA C (DNAJB1-protein kinase cAMP-activated catalytic subunit alpha); CIP4, CDC42-interacting protein 4; BC, breast cancer; CRC, colorectal cancer; GC, gastric cancer; HCC, hepatocellular carcinoma; HNSCC, head and neck squamous cell carcinoma; NSCLC, non-small-cell lung cancer; PC, prostate cancer, HCE, human corneal epithelial cell; HEK293, HEK cells. ↑ shows increased expression, and ↓ shows decreased expression of various targets.

**Table 2 cells-11-02020-t002:** Clinical trials on the cAMP–PKA pathway-targeting anticancer medicines (from clinicaltrials.gov (accessed on 15 June 2022)).

Identifier	Title	Cancer Type	Location
NCT00021268	Tocladesine in the treatment of progressive or recurrent metastatic colorectal cancer	Colorectal	Jonsson Comprehensive Cancer Center, UCLA Los Angeles, California, United States
NCT00004902	Tocladesine in the treatment of progressive or recurrent multiple myeloma	Multiple myeloma and plasma cell tumor	Robert H. Lurie Comprehensive Cancer Center, Northwestern University Chicago, Illinois, United States
NCT00004863	Paclitaxel and GEM 231 in the treatment of refractory or recurrent solid tumors	Unspecified adult solid tumor	Albert Einstein Comprehensive Cancer Center Bronx, New York, United States
NCT00004864	Docetaxel and GEM 231 in the treatment of refractory or recurrent solid tumors		

## Data Availability

Not applicable.

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
