# Peer review of "cAMP Signaling in Cancer: A PKA-CREB and EPAC-Centric Approach"

_cells, 2022, doi:10.3390/cells11132020_

Round 1
Reviewer 1 Report
please see the attachment.

Author Response
Reviewer 1 comments
Review of the manuscript: Cells-1744152. The review “cAMP signaling in cancer: a PKA CREB and EPAC-centric approach” submitted to Cells is a comprehensive and detailed summary of the role of cAMP signaling in cancer.
The authors have described numerous points of view regarding the effects of cAMP, PKA and Epac in tumor progression and metastasis, giving detailed information on the use of these proteins as potential anti-cancer therapeutic targets.
I have very few observations, that in my opinion need to be addressed: 1- In the introduction paragraph, lines 52-62: the authors should re-write the paragraph, because the message they are trying to communicate is not very clear. Please, try to fix the flow of the paragraph, to make it easier to read and understand.
Response:
Thankyou for your feedback. The paragraph has been revised. The flow of paragraph has also been maintained. Please refer to the Introduction section highlighted in green.
Comment 2
2- Paragraph 2, starting from line 106 to 159: The authors are claiming to describe the effects of cAMP/PKA signaling in several cancer type. However, they described extensively tumors related only to the brain. It will be very interesting if the authors will discuss also other cancer types, such as prostate, lung, ovaries and so on. There is a comprehensive review paper the authors can use, to find some good references (Zhang, H., Kong, Q., Wang, J. et al. Complex roles of cAMP–PKA–CREB signaling in cancer. Exp Hematol Oncol 9, 32 (2020). https://doi.org/10.1186/s40164-020-00191-1).
Response:
Thank you for your feedback. The section has been added with some recent data based on various cancers. Also, to enlist various cancer we added a table denoted as “table 1”. Please refer to the section 2 “The cAMP-PKA pathway's role in the growth of various tumors” highlighted in green and blue.
Comment 3
I have few minor comments: 1- Paragraph 3, line 163: “This happens when the G-coupled receptor tyrosine kinase (RTK) “. This is not correct. It is probably a typo error, but RTK receptor is not a G-coupled receptor. Please, clarify.
Response:
Thank you very much. The error has been corrected. Please refer to the section “Involvement of CREB in tumor growth” highlighted in green.
Comment 4
2- I invite the authors to revise the entire manuscript, for English errors and the reading flow. Some parts of the manuscript are very well written, but some others are difficult to read. Please, let somebody read the manuscript to fix any possible difficult sentences, to make them clearer and easier to understand.
Response:
Thank you for your feedback. The manuscript was revised and rewritten. Also, the manuscript was sent for English editing. We are happy to provide the English editing certificate if is required.
Reviewer 2 Report
This review by Ahmed et al. attempted to discuss the roles of cAMP signaling in cancer with a focus on the PKA-CREB and EPAC pathways. The manuscript has many issues. It is apparent that the authors are not very familiar with overall research fields, which is reflected by some of the inaccurate statements or errors that the authors made in the manuscript. Many of the statements and conclusions are not supported by necessary citations. Even when citations were present, they are often review papers instead of original studies. Moreover, the manuscript is not very up-to-date. Only a few citations are after 2019.
Specific issues:
· 1. Figure 1: PKA and Epac are parallel signaling pathways downstream of cAMP. As drawn now, the scheme is PKA centric with Epac downstream of PKA. Both the R and C subunits should be drawn to show the tetrameric structure of PKA. In the same Figure, CNG and HCN should be shown to be regulated by cAMP/cGMP. These channels are are non-selective cation channels that can conduct not just calcium but also other ions, and change the cell membrane potential.
· 2. In addition to compartmentalization by AKAP, PKA is also regulated in vivo by the differential expressions of the various R and C isoforms in a tissue/cell-specific and development-dependent manner.
· 3. Lines 84/85: “Epac proteins are polypeptides having catalytic and regulatory domains at the C-terminus.” This statement is not correct. The regulatory domain of the Epac is at the N-terminus while the catalytic component is at the C-terminal. Similarly, several statements in the same paragraph are not correct and/or accurate, the authors need to clean these errors up.
· 4. Many of the statements in the manuscripts are not backed up by proper citations. For example, in lines 80-81, “According to the tumor and its surrounding environment, the cAMP pathway and its downstream effectors may either inhibit or promote cancer.” Citations are needed to support the statement here. Also, lines 102-104: “Epac, a newly discovered cAMP effector, has a dual function in cancer, encouraging or inhibiting cancer formation and progression. As a result, Epac might be used as a cancer therapy target.” No citations are provided. There are a number of review articles on Epac’s roles in cancers, as well as applications of Epac modulators in various in vitro and in vivo models, have been published, these publications should be cited here.
· 5. Section 2 (line 105): “The cAMP-PKA pathway's role in the growth of various tumors”. This section is very shallow and narrowly focused. Some of the exciting progress that connects PKA mutations with fibrolamellar hepatocellular carcinoma and Cushing’s syndrome, as well as Carney complex were not discussed.
· 6. Section 3 (line 160). “Involvement of CREB in tumor growth”. Again, this section is very superficial and has very little association with the cAMP signaling.
· 7. Lines 195-197: “Epac stops the growth and survival of cancer cells in many types of cancer by targeting important signaling pathways involved in cell division, inflammation, and metabolic reprogramming.” Very vague description without detains and specific citations: What are the important signaling pathways involved?
· 8. Line 238: “Also, Epac breaks down X-ray repair cross-complementing protein 1 (XRCC1)” How does Epac break down XRCC1, Epac is not a protease?
· 9. Line 284: “While PKA catalytic subunits are mostly found in the mitochondrial outer membrane [112]”. Not sure this is correct. PKA C subunit is not known to mostly associate with mitochondrial out membrane.
· 10. Line 427: “CE3F4R, another Epac1 selective inhibitor, binds to the Epac1-cAMP complex non-competitively [160]” and Line 430: “When ESI-05 or ESI-07 binds to this domain, it stops Epac2 from becoming active [160].” Original references, not a review, should be cited. This is a pervasive problem throughout the manuscript.
· 11. Line 161: "8-pCPT-20-O-Me-cAMP [161]. 8-pCPT-20-O-Me-cAMP-AM” should be “8-pCPT-2’-O-Me-cAMP [161]. 8-pCPT-2’-O-Me-cAMP-AM”
· 12. PKA modulators had been explored extensively by the pharmaceutical industry in the 1980s and 1990s. Due to toxicity issues, they have not entered the clinics. Clinical trials listed in Table I were all quite old and started mostly more than a decade ago.
· 13. The style of the writing is on the casual side, not very scientific, and can be significantly improved. The authors should have their manuscript edited by a professional English proofreading service.
Author Response
Reviewer 2 comments:
This review by Ahmed et al. attempted to discuss the roles of cAMP signaling in cancer with a focus on the PKA-CREB and EPAC pathways. The manuscript has many issues. It is apparent that the authors are not very familiar with overall research fields, which is reflected by some of the inaccurate statements or errors that the authors made in the manuscript. Many of the statements and conclusions are not supported by necessary citations. Even when citations were present, they are often review papers instead of original studies. Moreover, the manuscript is not very up to date. Only a few citations are after 2019.
Response:
Thank you for your feedback. The whole manuscript has been reviewed and revised with new data and references. Most of the research shows review papers rather than original articles. The data on cAMP signaling in connection to various cancers is limited. However, recent data has been added. Please refer to the sections with highlighted colors, which show the updated work in the article. Also, proper citations have been cited.
Specific issues:
Comment 1:
Figure 1: PKA and Epac are parallel signaling pathways downstream of cAMP. As drawn now, the scheme is PKA centric with Epac downstream of PKA. Both the R and C subunits should be drawn to show the tetrameric structure of PKA. In the same Figure, CNG and HCN should be shown to be regulated by cAMP/cGMP. These channels are are non-selective cation channels that can conduct not just calcium but also other ions and change the cell membrane potential.
Response:
Thank you for your feedback. Please refer to the introduction section, the figure was revised.
Comment 2:
In addition to compartmentalization by AKAP, PKA is also regulated in vivo by the differential expressions of the various R and C isoforms in a tissue/cell-specific and development-dependent manner.
Response:
Thank you for your feedback. We agree with you.
Comment 3:
Lines 84/85: “Epac proteins are polypeptides having catalytic and regulatory domains at the C-terminus.” This statement is not correct. The regulatory domain of the Epac is at the N-terminus while the catalytic component is at the C-terminal. Similarly, several statements in the same paragraph are not correct and/or accurate, the authors need to clean these errors up.
Response:
Thank you for your feedback. The error has been corrected. Also, a new figure (Figure 2) has been added which shows the structure and mechanism of these domains. Please refer to the “Introduction section” highlighted in green.
Comment 4:
Many of the statements in the manuscripts are not backed up by proper citations. For example, in lines 80-81, “According to the tumor and its surrounding environment, the cAMP pathway and its downstream effectors may either inhibit or promote cancer.” Citations are needed to support the statement here. Also, lines 102-104: “Epac, a newly discovered cAMP effector, has a dual function in cancer, encouraging or inhibiting cancer formation and progression. As a result, Epac might be used as a cancer therapy target.” No citations are provided. There are several review articles on Epac’s roles in cancers, as well as applications of Epac modulators in various in vitro and in vivo models, have been published, these publications should be cited here.
Response:
Thank you for your feedback. Citations have been added. Please refer to the “Introduction section” highlighted in yellow.
Comment 5:
Section 2 (line 105): “The cAMP-PKA pathway's role in the growth of various tumors”. This section is very shallow and narrowly focused. Some of the exciting progress that connects PKA mutations with fibrolamellar hepatocellular carcinoma and Cushing’s syndrome, as well as Carney complex were not discussed.
Response:
Thank you for your feedback. The section 2 “The cAMP-PKA pathway's role in the growth of various tumors” has been revised and a new table 1 has been added showing various cancers and other related effectors to cAMP signaling. Also, Cushing’s syndrome, as well as Carney complex related data have been added. Please refer to the section 2 “The cAMP-PKA pathway's role in the growth of various tumors” highlighted in green and blue.
Comment 6:
Section 3 (line 160). “Involvement of CREB in tumor growth”. Again, this section is very superficial and has very little association with the cAMP signaling.
Response:
Thank you for your feedback. The section has been revised with some more data. Please refer to the section 3 “Involvement of CREB in tumor growth” highlighted in green.
Comment 7:
Lines 195-197: “Epac stops the growth and survival of cancer cells in many types of cancer by targeting important signaling pathways involved in cell division, inflammation, and metabolic reprogramming.” Very vague description without detains and specific citations: What are the important signaling pathways involved?
Response:
Thank you for your feedback. This section has been revised and some more data has been added. Please refer to the section 4 “Involvement of EPAC in tumor growth” highlighted in green.
Comment 8:
Line 238: “Also, Epac breaks down X-ray repair cross-complementing protein 1 (XRCC1)” How does Epac break down XRCC1, Epac is not a protease?
Response:
Thank you for your feedback. The actual sentence was “Epac has been shown to mediate cAMP-induced inhibition of DNA damage repair in lung cancer by promoting the degradation of X-ray repair cross-complementing protein 1 (XRCC1)” The sentence has been removed and the data has been revised. Please refer to the section 4 “Involvement of EPAC in tumor growth” highlighted in green.
Comment 9:
Line 284: “While PKA catalytic subunits are mostly found in the mitochondrial outer membrane [112]”. Not sure this is correct. PKA C subunit is not known to mostly associate with mitochondrial out membrane.
Response:
Thank you for your feedback. The research says “PKA components have been found in the mitochondria of some species and tissues, for example, in the mouse oocyte”. Please for further clearance refer to the following article entitled as “Protein Kinase A Subunit α Catalytic and A Kinase Anchoring Protein 79 in Human Placental Mitochondria “. However, to avoid confusion, the whole section has been revised with new data. Please refer to the section 5” cAMP and its other effectors act in various signaling pathways” highlighted in green.
Comment 10:
Line 427: “CE3F4R, another Epac1 selective inhibitor, binds to the Epac1-cAMP complex non-competitively [160]” and Line 430: “When ESI-05 or ESI-07 binds to this domain, it stops Epac2 from becoming active [160].” Original references, not a review, should be cited. This is a pervasive problem throughout the manuscript.
Response:
Thank you for your feedback. Original research has been cited. Please refer to the section 6 potential anti-cancer therapeutic strategies” highlighted in yellow.
Comment 11:
Line 161: "8-pCPT-20-O-Me-cAMP [161]. 8-pCPT-20-O-Me-cAMP-AM” should be “8-pCPT-2’-O-Me-cAMP [161]. 8-pCPT-2’-O-Me-cAMP-AM”
Response:
Thank you for your feedback. The correction has been made. Please refer to the section 6 potential anti-cancer therapeutic strategies” highlighted in yellow.
Comment 12:
PKA modulators had been explored extensively by the pharmaceutical industry in the 1980s and 1990s. Due to toxicity issues, they have not entered the clinics. Clinical trials listed in Table I were all quite old and started mostly more than a decade ago.
Response:
Thank you for your feedback. Yes, we agree with you, but to our knowledge there is no updated data on clinical trials which makes it more important that this issue must be brought up in the future.
Comment 13:
The style of the writing is on the casual side, not very scientific, and can be significantly improved. The authors should have their manuscript edited by a professional English proofreading service
Response:
Thank you for your feedback. The whole article has been revised; the style has been adjusted by the journal, for which we are thankful; and the article has been sent for editing. An editing certificate can be provided if required.
Reviewer 3 Report
The paper submitted for review is constructed with several subsections and looks like a systematic review, but it is a narrative paper. There is no information about the purpose of the paper, neither in the abstract nor in the Introduction section. The biggest weakness of the paper, however, is the old literature. Only 23/181 (about 13%) of the cited papers are from the last 5 years. Obviously the key papers in this topic need to be cited, but not in every subsection. The paper is difficult to read because it contains many mental shortcuts, information "cut" as if from large papers, not always related to the topic of carcinogenesis (lines 281-308). The authors' own comments are missing.
Other comments:
1. since the authors have done a tremendous amount of work, I recommend improving the Methods section: what keywords were used to search the literature and describe the cAMP signaling system in cancer. Since the general knowledge is very broadly already described in the literature, please narrow it down to cancer only.
Descriptions of all figures should be standardized, with appropriate explanations of abbreviations, in small or capital letters - see Epac - line 271 and EPAC - line 275; add a short description of the function of a given pathway or effector enzyme/protein, e.g. compare the description of Figure 1 and 3.
Figure 1 - I recommend extending the legend and adding an explanation of some abbreviations, e.g. AKAP, EPAC, PDE, etc. write where the data comes from - is it author's or another publication.
2. lines 83-104, such description in the text is illegible, maybe a figure here?
3. subsection 2 - it would be advisable to include a table as it seems that only a few cancers are presented in it, is cAMP pathway important only in these cancers; please add a summary of this part.
4. why is Epac so prominent in prostate cancer? Is it the same in other cancers? please also throw out the repeated text : lines 201-203 and 208-210 - same text.
5. subsection 5 - lines 281-308 there is no knowledge about cancer, is this data necessary? I find incomprehensible these divagations of Epac and ROS, please shorten.
6. Table 1 - data is based on clinicaltrials.gov, please cite this website in the literature.
7. conclusions need to be based on the paper as a whole, what is really new that the authors show, what conclusions do the papers presented lead to, what does the latest research on this signaling pathway in cancers really contribute, etc.
8. I recommend reviewing the entire paper for an explanation of abbreviations used for the first time, starting with PKA in the paper's abstract (line 19).
9. Please rewrite the literature as recommended by the journal "Cells" as I do not think it is correct.
10. If the paper is to be accepted for publication, absolutely must add newer papers at least 10% and modified chapters.
Author Response
Reviewer 3
Comments
The paper submitted for review is constructed with several subsections and looks like a systematic review, but it is a narrative paper. There is no information about the purpose of the paper, neither in the abstract nor in the Introduction section. The biggest weakness of the paper, however, is the old literature. Only 23/181 (about 13%) of the cited papers are from the last 5 years. Obviously, the key papers in this topic need to be cited, but not in every subsection. The paper is difficult to read because it contains many mental shortcuts, information "cut" as if from large papers, not always related to the topic of carcinogenesis (lines 281-308). The authors' own comments are missing.
Response:
Thank you for your feedback. The purpose of paper has been added. Please refer to “abstract” highlighted in blue. The literature has been updated and recent references have been added which are almost 100/279. Beyond this, to our knowledge, there is no more recent data available showing cAMP signaling and its relation to cancer. Please refer to the section “reference” highlighted in yellow. The manuscript was revised and rewritten. Also, the manuscript was sent for English editing. We are happy to provide the English editing certificate if is required.
Other comments:
Since the authors have done a tremendous amount of work, I recommend improving the Methods section: what keywords were used to search the literature and describe the cAMP signaling system in cancer. Since the general knowledge is very broadly already described in the literature, please narrow it down to cancer only.
Response:
Thank you for your feedback. Yes, we agree with you. The literature has been revised and to our best it is limited to the cAMP and its effectors signaling system in cancer.
Descriptions of all figures should be standardized, with appropriate explanations of abbreviations, in small or capital letters - see Epac - line 271 and EPAC - line 275; add a short description of the function of a given pathway or effector enzyme/protein, e.g., compare the description of Figure 1 and 3.
Response:
Thank you for your feedback. The descriptions have been standardized. Also, the text includes the full descriptions.
Figure 1 - I recommend extending the legend and adding an explanation of some abbreviations, e.g., AKAP, EPAC, PDE, etc. write where the data comes from - is it author's or another publication.
Response:
Thank you for your feedback. That figure has been revised and reference has been mentioned, the abbreviations are updated.
- lines 83-104, such description in the text is illegible, maybe a figure here?
Response:
Thank you for your feedback. A new figure has been added. Please refer to figure 2.
- subsection 2 - it would be advisable to include a table as it seems that only a few cancers are presented in it, is cAMP pathway important only in these cancers; please add a summary of this part.
Response:
Thank you for your feedback. A new table has been added. Please refer to table 1. Also, some more data has been added to this section. Please refer to section 2” The cAMP-PKA pathway's role in the growth of various tumors” highlighted in green and blue.
- why is Epac so prominent in prostate cancer? Is it the same in other cancers? please also throw out the repeated text: lines 201-203 and 208-210 - same text.
Response:
Thank you for your feedback. Perhaps, but we presented EPAC not only in prostate. Figure 5 also shows the role of EPAC in lymphocytic leukemia. Plus, some more data has been added. The repeated text has been removed highlighted in grey. Please refer to section 4” Involvement of EPAC in tumor growth” highlighted in green and grey.
- subsection 5 - lines 281-308 there is no knowledge about cancer, is this data necessary? I find incomprehensible these divagations of Epac and ROS, please shorten.
Response:
Thank you for your feedback. Data has been revised and divagations of EPAC and ROS has been removed to avoid confusion. Plus, some data has been added. Please refer to section 4” Involvement of EPAC in tumor growth” highlighted in green.
- Table 1 - data is based on clinicaltrials.gov, please cite this website in the literature.
Response:
Thank you for your feedback. The website has been cited.
- conclusions need to be based on the paper, what is really new that the authors show, what conclusions do the papers presented lead to, what does the latest research on this signaling pathway in cancers really contribute, etc.
Response:
Thank you for your feedback. The conclusion has been added with some more remarks. Please refer to conclusion highlighted in blue.
- I recommend reviewing the entire paper for an explanation of abbreviations used for the first time, starting with PKA in the paper's abstract (line 19).
Response:
Thank you for your feedback. The paper has been reviewed thoroughly. The abbreviations almost been corrected.
- Please rewrite the literature as recommended by the journal "Cells" as I do not think it is correct.
Response:
Thank you for your feedback. The paper has been rewritten.
- If the paper is to be accepted for publication, absolutely must add newer papers at least 10% and modified chapters.
Response:
Thank you for your feedback. Many new and recent references have been added. Please refer to sections highlighted in green, yellow, and blue which consist mostly recent research.
Reviewer 4 Report
cAMP may have tumor-promoting or tumor-suppressive roles depending on the context and tumor types. It may be better if there is a table or figure to show the contrasting roles of cAMP signaling in different types of cancer and discuss the potential mechanisms of these effects.
Author Response
Comments
cAMP may have tumor-promoting or tumor-suppressive roles depending on the context and tumor types. It may be better if there is a table or figure to show the contrasting roles of cAMP signaling in different types of cancer and discuss the potential mechanisms of these effects.
Response:
Thank you for your feedback. A new table has been added. Please refer to table 1. Also, some more data has been added to this section. Please refer to section 2” The cAMP-PKA pathway's role in the growth of various tumors” highlighted in green and blue.
Round 2
Reviewer 3 Report
The authors exhaustively responded to my doubts and questions posed by myself following reading of the first version of the paper. Most of the data or corrections have been introduced everywhere where they were required.
Nevertheless, I still have comments on the newly created Table 1.
Table 1 is a bit incomprehensible to me, what is the key to understanding how the cAMP/PKA system and POPDC proteins work? There are some cancers listed in the first column, and then in the column next to the proteins, other cancers. What are the signaling pathways involved? I don't know what down- and up-regulations refer to?
Table should be slightly reworded, moreoverit should be placed further down in the text where it is quoted. Please complete the explanations of some abbreviations, especially in the heading, even if they are explained earlier. All explanations, including increases and decreases in regulation and letter abbreviations, should be below the table.
I think that after these small corrections, the paper can be accepted for printing in the current version.
Author Response
Reviewer 3 comments (2)
The authors exhaustively responded to my doubts and questions posed by myself following reading of the first version of the paper. Most of the data or corrections have been introduced everywhere where they were required.
Nevertheless, I still have comments on the newly created Table 1.
Table 1 is a bit incomprehensible to me, what is the key to understanding how the cAMP/PKA system and POPDC proteins work? There are some cancers listed in the first column, and then in the column next to the proteins, other cancers. What are the signaling pathways involved? I don't know what down- and up-regulations refer to?
Table should be slightly reworded, moreover it should be placed further down in the text where it is quoted. Please complete the explanations of some abbreviations, especially in the heading, even if they are explained earlier. All explanations, including increases and decreases in regulation and letter abbreviations, should be below the table.
I think that after these small corrections, the paper can be accepted for printing in the current version.
Response (2):
Thank you for your feedback. The due responses are mentioned below.
How the cAMP/PKA system and POPDC proteins work?
1.Working mechanism:
Popeye domain-containing proteins (POPDC) are a fourth class of cAMP-binding proteins. In terms of structure, the Popeye domain is quite like cAMP-binding proteins. Structures from CRP and PKA were used to develop a homology model. This model showed that the putative cAMP-binding domain has a substantial number of invariant amino acids clustered around it, indicating that these amino acids are involved in nucleoside binding. cAMP binding was shown to be dependent on residues that were altered using mutagenesis.
The 150-amino acid Popeye domain is considered to primarily provide the function of binding cAMP. The Popeye domain's predicted structure resembled the CNBD of PKA's catalytic subunit. As with many proteins, the CNBD structure is classified as a -barrel fold. Non-cyclic nucleotides may be bound by some of these proteins as well. One aspect of the CNBD that hasn't changed over time is the phosphate-binding cassette (PBC), which is situated between sheets 6 and 7 of the -helix and makes direct contact with cAMP. Two conserved residues can be identified in all PBCs: an arginine that binds to the phosphate group of cAMP and a glutamate that binds to the 2′-OH group of ribose. Other ligands besides cNMPs may bind to proteins that have the jellyroll-barrel fold structure but have no arginine or glutamate residues as a result, it is imperative that adequate experimental data be provided so that the binding of cAMP to POPDC proteins may be demonstrated definitively. Two conserved sequence motifs (FL/IDSPEW/F and FQVT/S) are connected by a non-conserved sequence of varying length in the non-canonical PBC of the Popeye domain. By mutating these evolutionarily conserved residues using a charge-to-alanine method in POPDC1 and POPDC2, we were able to reduce the binding affinity of POPDC1 and POPDC2, which supports their role in nucleotide binding.
Tests for cAMP binding by POPDC proteins were devised. CAMP agarose precipitation assays were used to confirm the binding of POPDC1 to cAMP in chicken heart extracts. CGMP and free cGMP were able to remove POPDC1 from POPDC1-agarose, therefore it remained unclear if POPDC1 binds cAMP preferentially, or both cyclic nucleotides equally, at this point. All three POPDC proteins were shown to bind to cAMP in the agarose precipitation assay. POPDC1 was recombinantly expressed and used in a radio-ligand binding test to determine an IC50 value of 118.4 ± 7.1nM for cAMP and 5.27 ± 0.68μM for cGMP, respectively. Similar variances in affinity for both nucleotides have been found in the CNBDs of PKA's cAMP-binding proteins, such as cAMP-binding proteins (CBPs).
Based on POPDC1 and KCNK2 interactions, another test for cAMP binding has been created (TREK-1). TREK-1 and POPDC1 were both fluorescently labeled with YFP to perform a bimolecular FRET experiment. After adding isoproterenol or forskolin, both of which act directly on Adenylate Cyclase (adenosine monophosphate), the YFP/CFP ratio dropped rapidly, following the normal kinetics of an Adenosine Cyclase-dependent process. After the addition of isoproterenol, a mutant protein with a mutation in the DSPE motif residue D200 in the PBC of POPDC1 had no effect on the FRET signal. Also, nitroprusside, which increases the quantity of cGMP in cells, had no effect on the FRET signal, which supports the concept that POPDC1 protein binds to cAMP but not cGMP at physiological quantities.
Finally, the finding of a POPDC1S201F mutation in individuals with limb-girdle muscular dystrophy (LGMD) phenotype provides support for the binding of cAMP to POPDC proteins. cAMP-binding affinity is anticipated to be reduced by the serine to phenylalanine substitution at POPDC1 serine residue 201. A 50% drop in cAMP affinity was seen in the mutant protein, as measured by cAMP affinity assays. Thus, the Popeye domain has been established as a unique cAMP-binding domain with a highly varied protein sequence by a wide range of experiments and genetic data. However, so far, the molecular characterization has focused on the POPDC1 protein, and thus it is important to extend these experiments to the other two POPDC proteins and determine whether cyclic nucleotide specificity is the same for each of the POPDC proteins, and whether all three POPDC isoforms bind cAMP with equal affinity.
- What are the signaling pathways involved?
Please refer to the table 1. It has been updated with the recently mentioned signaling pathways involved in cancer, which are highlighted in "yellow" color.
- I don't know what down- and up-regulations refer to?
To avoid confusion, the concept has been changed. Please refer to table 1, highlighted in "yellow" color.
- Table should be slightly reworded, moreover it should be placed further down in the text where it is quoted. Please complete the explanations of some abbreviations, especially in the heading, even if they are explained earlier. All explanations, including increases and decreases in regulation and letter abbreviations, should be below the table.
Table is reworded and added with some more data. Also, the table has been shifted down in the text. Explanations and some abbreviations have been updated. All explanations, including increases and decreases in regulation and letter abbreviations, are placed below the table. Please refer to table 1, highlighted in "yellow" color.